# Knowledge of Diabetes among Adults at High Risk for Type 2 Diabetes in the Trivandrum District of Kerala, India

**Thirunavukkarasu Sathish** [1,*] , **Kavumpurathu Raman Thankappan** [2] , **Jeemon Panniyammakal** [3] **and Brian Oldenburg** [4]

1   Department of Family and Preventive Medicine, School of Medicine, Emory University, Atlanta, GA 30322, USA
2   Department of Public Health, Amrita Institute of Medical Sciences, Kochi 682 041, Kerala, India
3   Achutha Menon Centre for Health Science Studies, Sree Chitra Tirunal Institute for Medical Sciences and Technology, Trivandrum 695 011, Kerala, India
4   Baker Heart and Diabetes Institute, Melbourne, VIC 3004, Australia
*   Correspondence: sathish.thirunavukkarasu@emory.edu; Tel.: +1-(470)-357-8308

**Abstract:** We aimed to study the knowledge of diabetes among individuals with a high risk for developing type 2 diabetes in the Trivandrum district of the Indian state of Kerala. The baseline data collected from 1007 participants of the Kerala Diabetes Prevention Program were analyzed. Diabetes knowledge was assessed using a scale adapted from a large nationwide study conducted in India. The composite score of the scale ranged from 0 to 8. The mean age of the participants was 46.0 (SD: 7.5) years, and 47.2% were women. The mean diabetes knowledge score was 6.9 (SD: 2.1), with 59.5% having the maximum possible score of 8. Of the 1007 participants, 968 (96.1%) had heard the term diabetes, and of them, 87.2% knew that the prevalence of diabetes is increasing, 92.9% knew at least one risk factor for diabetes, 79.6% knew that diabetes can cause complications in organs, and 75.9% knew that diabetes can be prevented. While the overall level of knowledge of diabetes about its risk factors, complications, and prevention was generally high, an alarmingly low proportion of participants knew that diabetes can affect key organs such as the eyes (24.0%), heart (20.1%), feet (10.2%), and nerves (2.9%), and nearly a quarter (24.1%) were not aware that diabetes can be prevented. It is essential to educate high-risk individuals about diabetes complications and the importance of and strategies for diabetes prevention in the Trivandrum district of Kerala.

**Keywords:** diabetes knowledge; diabetes; prediabetes; prevention; health promotion; awareness

## 1. Introduction

Diabetes imposes significant health and economic challenges globally, especially in low- and middle-income countries (LMICs) such as India [1]. An estimated 537 million adults (20–79 years) were living with diabetes worldwide in 2021, of which 14% (74.2 million) were from India, the country with the second largest number of people with diabetes in the world. This number is projected to increase by 68% (124.9 million) over the next 25 years. More worryingly, slightly more than half (53.1%) of people with diabetes in India are unaware of their condition [1], putting them at risk of developing complications. Furthermore, India has the third highest annual number of deaths from diabetes globally, at about 0.6 million [1]. On the economic side, a 2020 systematic review of 32 studies showed that the median direct cost of diabetes was INR 21,000 (USD ~255) in India [2]. A substantial proportion (70%) of these costs was incurred by patients through out-of-pocket spending. The above figures underscore the importance of the prevention of diabetes to control the health and economic burden of diabetes in India.

Efficacy trials and implementation studies have clearly and consistently shown that lifestyle interventions incorporating physical activity and healthy diets can reduce diabetes incidence in high-risk South Asians (e.g., Indians) [3]. However, the individual

and community-level uptake and the data on long-term outcomes of diabetes prevention programs have been very limited in India and other South Asian countries [3]. These problems are mainly attributed to poor political commitment, limited resources, poor resource allocation, competing priorities (e.g., infectious diseases), and low levels of awareness and knowledge about diabetes in general and its risk factors, complications, and prevention strategies in these countries [4].

While it is essential to know the level of awareness of types 1 and 2 diabetes in the community, when it comes to prevention, type 2 diabetes gains the utmost importance. Indeed, studies suggest that increasing the knowledge of diabetes and its risk factors among the population will likely have a substantial benefit in preventing type 2 diabetes [5,6]. Considering the increasing burden of type 2 diabetes in India, it is imperative to understand the knowledge of diabetes risk factors, complications, and prevention to design culturally appropriate educational strategies. Studies conducted in India have examined the diabetes knowledge of the general population [5,7–12] and people with diabetes [10,13]. These studies have shown that the knowledge about various aspects of diabetes was generally low among the studied populations [5,7–13]. However, to the best of our knowledge, there are no studies among those at high risk of developing diabetes in India. It is essential to understand the level of diabetes knowledge in the high-risk group for diabetes, as increasing it could help prevent diabetes among these individuals [5,6]. Therefore, we aimed to study the knowledge of diabetes among high-risk individuals for type 2 diabetes in the Trivandrum district of the Indian state of Kerala.

## 2. Materials and Methods

### 2.1. Study Design and Study Population

For this analysis, we used the baseline data of the participants of the Kerala Diabetes Prevention Program (K-DPP), a lifestyle-based diabetes prevention cluster-randomized controlled trial conducted in Kerala state [14]. The K-DPP study design and its screening and recruitment have been described in detail elsewhere [14–16]. Figure 1 shows the screening and recruitment of K-DPP participants. Briefly, we randomly selected 60 polling areas (electoral divisions with geographical boundaries) from 359 polling areas in a *taluk* (the unit below the district) of the Trivandrum district, Kerala. From the electoral roll of these 60 polling areas (clusters), which were equally randomized to intervention and control groups, we randomly identified individuals aged 30 to 60 years for screening. These individuals were approached at their households for phase 1 screening, which involved the administration of a questionnaire with the eligibility criteria and the Indian Diabetes Risk Score (IDRS) [17]. The potentially eligible individuals (1) should be literate in Malayalam (the local language); (2) should have no history of diabetes, heart disease, stroke, cancer, epilepsy, arthritis, or dementia; (3) should not be pregnant; and (4) should not be currently taking glucose-lowering medications (e.g., steroids, antipsychotics). Those satisfying the eligibility criteria were screened with the IDRS, which is composed of age, family history of diabetes, physical activity, and waist circumference [17]. The IDRS has been widely validated across several states in India among a variety of populations (e.g., high-risk people for diabetes, diabetes patients) [18] and is a strong predictor of incident type 2 diabetes [19,20]. Individuals with an IDRS of ≥60 [17] were considered to be at high risk of developing diabetes and were subsequently invited to attend community-based clinics for confirmation with a 2-hour 75 g oral glucose tolerance test (OGTT) (phase 2 screening). Those diagnosed with diabetes (fasting plasma glucose (FPG) ≥ 126 mg/dL or 2-hour plasma glucose (2-hr PG) ≥ 200 mg/dL) [21] on the OGTT were excluded from further participation in the study and referred to healthcare centers for treatment and care. The remaining individuals with normal glucose tolerance (FPG < 100 mg/dL and 2-hr PG < 140 mg/dL) [21] and prediabetes (FPG 100 to 125 mg/dL or 2-hr PG 140 to 199 mg/dL) [21] were included in the trial. After screening 3689 adults in two phases, we identified 1007 high-risk individuals and 202 people with diabetes. We considered the 1007 high-risk individuals for this analysis.

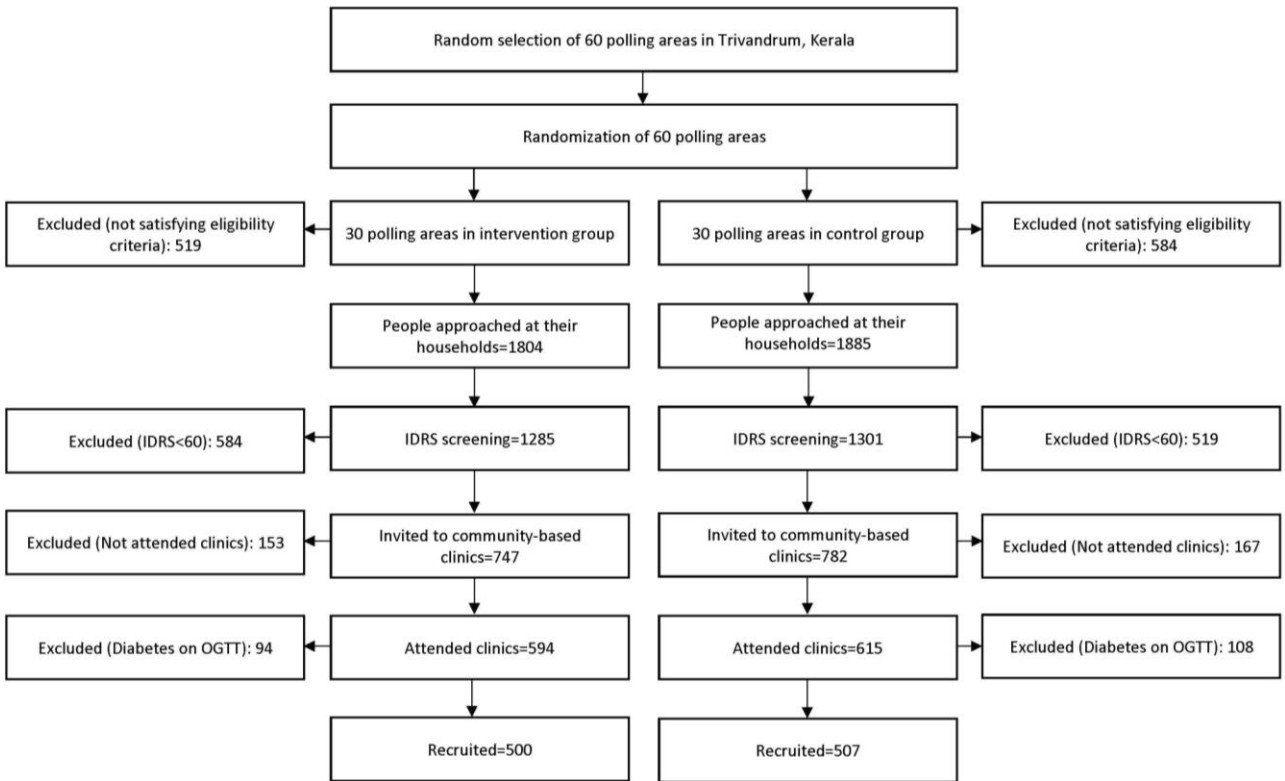

**Figure 1.** Flowchart showing the screening and recruitment of K-DPP participants. IDRS, Indian Diabetes Risk Score; OGTT, oral glucose tolerance test.

*2.2. Diabetes Knowledge Scale*

We collected data on the knowledge of diabetes using a scale developed in the Indian Council of Medical Research–India Diabetes (ICMR-INDIAB) study (Phase I), a large cross-sectional study on diabetes, involving representative samples from four states in India, one from the North, South, East, and West, covering a population of 213 million [7]. This scale includes the questions below about a person's knowledge of diabetes concerning its burden, risk factors, complications, and prevention:

1. Have you heard of the term diabetes? Yes/No
2. If yes, do you think, in general, more and more people are getting affected with diabetes nowadays? Yes/No/Don't know
3. What are the factors you think contribute to diabetes? (please circle as many as appropriate)
   a. Obesity
   b. Decreased physical activity
   c. Family history of diabetes
   d. Mental stress
   e. Unhealthy diet
   f. Tobacco use
   g. Alcohol use
   h. Others (specify)
   i. Don't know
4. Do you know that diabetes can cause complications in organs? Yes/No/Don't know
5. If yes, which organs are they?
   a. Eyes
   b. Heart
   c. Nerves
   d. Feet

    e.    Kidneys
    f.    Hands and Fingers
    g.    Bones
    h.    Brain
    i.    Liver
    j.    Skin
    k.    Others (specify)
    l.    Don't know
6.    Can diabetes be prevented? Yes/No/Don't know

We computed a composite score for the knowledge of diabetes as described in previous studies [7,8] and as follows: (1) For closed questions, correct answers were given a score of 1 and incorrect answers (including "don't know") a score of 0. (2) For risk factors for diabetes, a score of 4 was assigned for participants who answered obesity, unhealthy diet, physical inactivity, or family history of diabetes; 3 for consuming sweets; 2 for mental stress; 1 for tobacco or alcohol use; and 0 for don't know. The least possible score was 0, and the maximum score was 8. Data to calculate the composite diabetes knowledge score were available for 998 (99.1%) of the participants. The scale showed excellent internal consistency with this sample (Cronbach's $\alpha$ = 0.81) [22].

### 2.3. Statistical Analysis

Data were examined for normality using the Shapiro–Wilk test and histograms. According to the nature of the distribution, continuous variables are summarized using mean and standard deviation (SD) or median and interquartile range (IQR), and categorical variables with frequency and percentage. The internal consistency of the scale was estimated using the "alpha" command in STATA software (Version 17.0, StataCorp, College Station, TX, USA).

## 3. Results

### 3.1. Characteristics of Participants

A total of 1007 high-risk individuals were included in this analysis. The mean age was 46.0 (SD: 7.5) years, and 47.2% were women. The majority were educated up to higher secondary school (90.0%), employed (72.3%), and married (95.1%) (Table 1). The median monthly household expenditure was INR 7000 (IQR: 5000–10,000).

**Table 1.** Sociodemographic characteristics of the participants.

| Characteristics | N = 1007 |
|---|---|
| Age (years), mean (SD) | 46.0 (7.5) |
| Sex, n (%) | |
| Men | 532 (52.8) |
| Women | 475 (47.2) |
| Education, n (%) | |
| Up to primary | 253 (25.1) |
| Above primary to higher secondary school | 653 (64.9) |
| College or above | 101 (10.0) |
| Occupation, n (%) | |
| Skilled/unskilled | 728 (72.3) |
| Homemaker | 268 (26.6) |
| Unemployed/retired | 11 (1.1) |

**Table 1.** *Cont.*

| Characteristics | N = 1007 |
|---|---|
| Marital status, n (%) | |
| Single | 11 (1.1) |
| Married | 958 (95.1) |
| Divorced/separated/widowed | 38 (3.8) |
| Monthly household expenditure (INR), median (IQR) | 7000 (5000–10,000) |

SD, standard deviation; IQR, interquartile range; INR, Indian rupees.

*3.2. Components of Diabetes Knowledge Scale and Diabetes Knowledge Score*

Of 1007 participants, 968 (96.1%) had heard the term diabetes. Of them, 87.2% thought that more and more people are getting diabetes nowadays (i.e., diabetes prevalence is increasing), 92.9% knew at least one risk factor for diabetes, 79.6% knew that diabetes can cause complications in organs, and 75.9% knew that diabetes can be prevented (Figure 2). Family history of diabetes was the most commonly reported risk factor (78.7%), followed by unhealthy diet (70.1%), physical inactivity (64.2%), obesity (62.5%), alcohol use (44.3%), stress (41.0%), tobacco use (34.5%), don't know (6.7%), and taking certain types of medications (0.3%) (Figure 3). The kidney was the most commonly reported organ (42.8%) affected by diabetes, followed by eyes (24.0%), liver (23.5%), heart (20.1%), don't know (18.4%), feet (10.2%), hands and fingers (8.1%), brain (3.4%), nerves (2.9%), others (2.9%), bones (2.8%), and skin (0.5%) (Figure 4). The mean diabetes knowledge score was 6.9 (SD: 2.1), with 59.5% having the maximum possible score of 8 and 3.9% having the least possible score of 0.

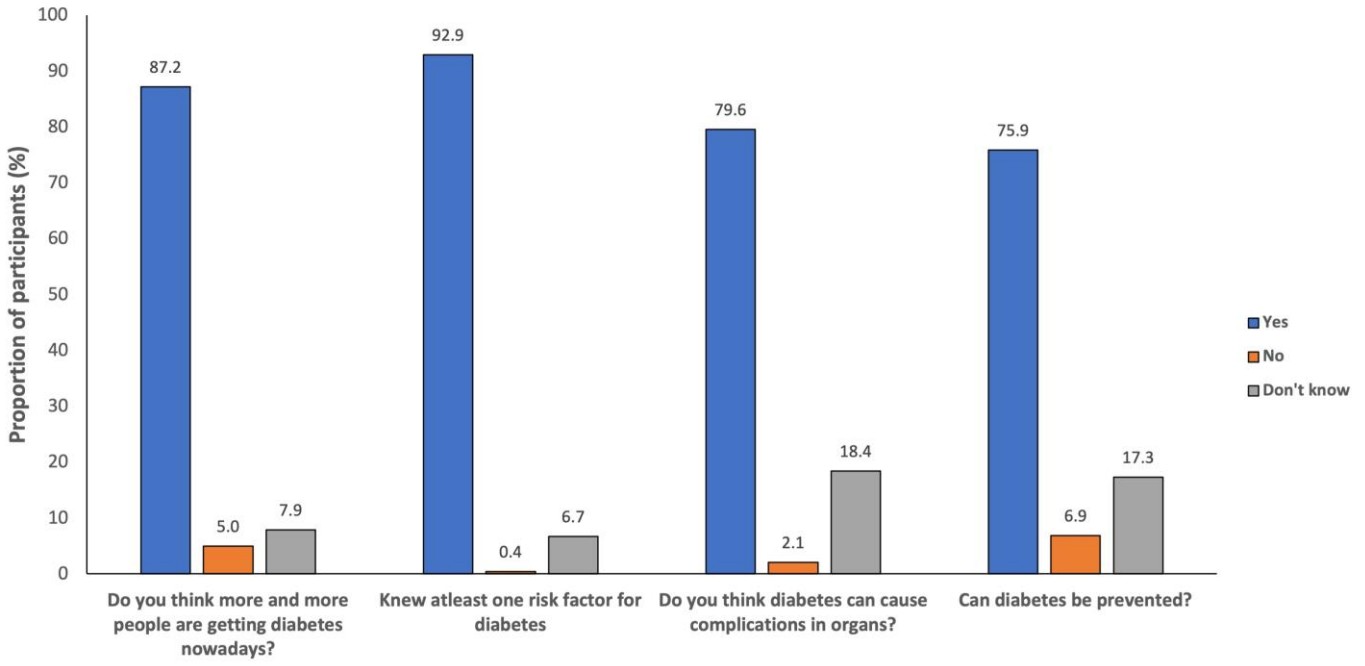

**Figure 2.** Components of diabetes knowledge scale among adults at high risk for type 2 diabetes.

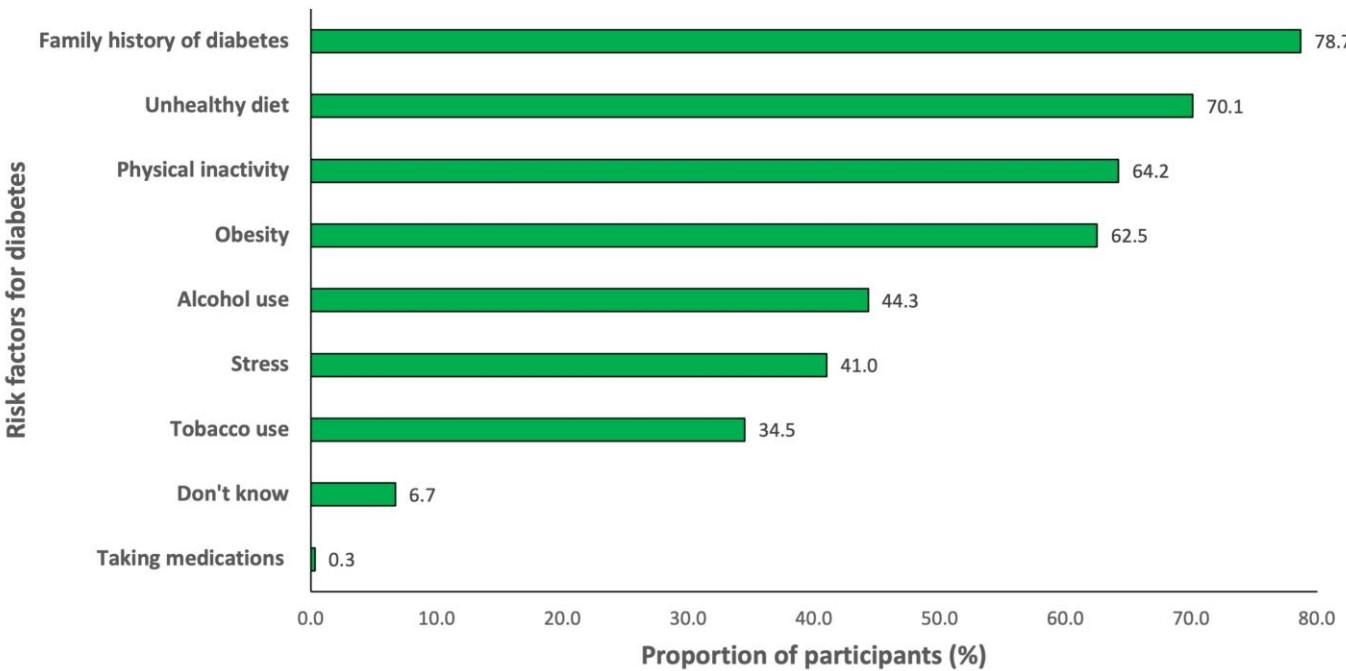

**Figure 3.** Knowledge of risk factors for type 2 diabetes.

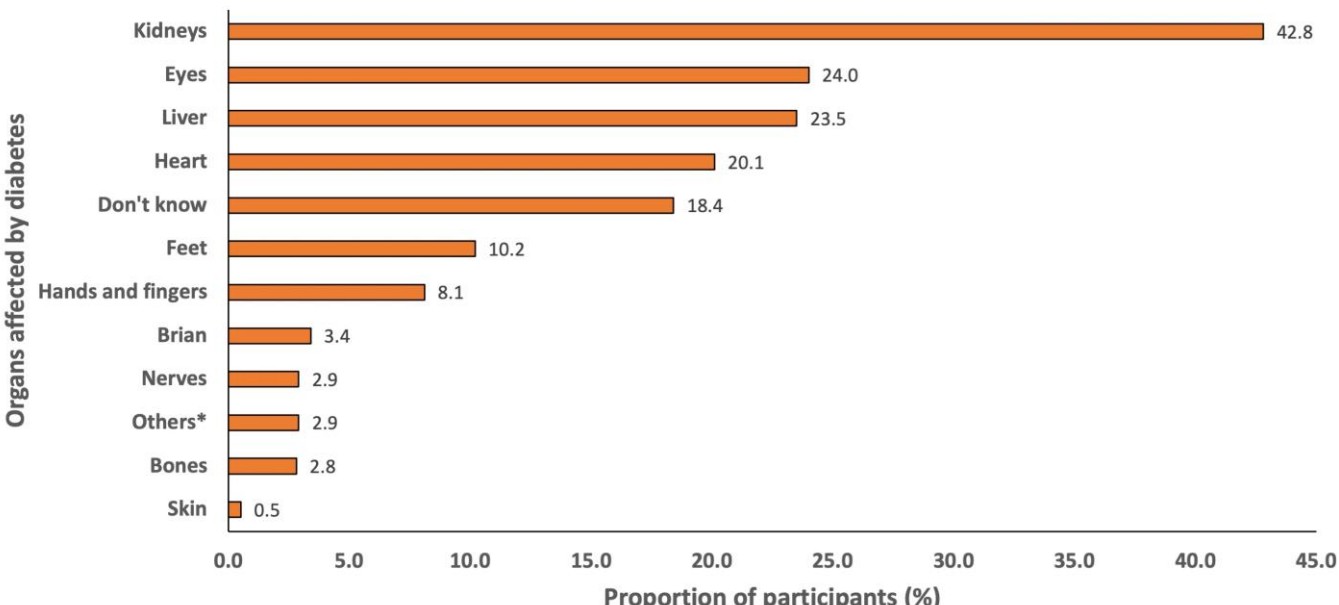

**Figure 4.** Knowledge of organs affected by type 2 diabetes. * Others include sexual organs, blood vessels, lungs, stomach, intestine, teeth, and intestine.

### 4. Discussion

This is the first study from India to assess the knowledge of diabetes concerning its risk factors, complications, and prevention among individuals at high risk of developing type 2 diabetes. The overall level of knowledge of our study participants about the various components of diabetes was high, with proportions ranging from 75.9% to 92.9%. However, alarmingly low proportions were aware that diabetes affects certain key organs, including the eyes (24.0%), heart (20.1%), feet (10.2%), and nerves (2.9%), and nearly a quarter (24.1%) of participants did not know that diabetes can be prevented.

Given the high-risk nature of our study participants for developing diabetes, as expected, the level of knowledge concerning various components of diabetes was higher compared with that from studies conducted among the general population in India [5,7–11]. For example, in the ICMR-INDIAB study (Phase I) [7], which included 13,794 adults (≥20 years) from the general population, 41.5% had heard the term diabetes, 80.7% were aware that the prevalence of diabetes is increasing these days, 51.4% knew that diabetes could cause complications in organs, and 56.3% knew that diabetes can be prevented [7]. The corresponding figures in our study were 96.1%, 87.2%, 79.6%, and 75.9%, respectively. However, surprisingly, in the ICMR-INDIAB study, among those with self-reported diabetes (n = 480), only 72.7% knew that diabetes could affect organs, and 63.4% were aware that prevention of diabetes is possible [7], which were lower than those in our study (79.6% and 75.9%, respectively). In another study conducted among 385 patients with type 2 diabetes in three government hospitals in Delhi, the average composite diabetes knowledge score was 3.8 out of a maximum possible score of 7 (54.3%) [13]. In our study, the score was higher at 6.9 out of 8 (86.3%). These differences in findings between our study and other studies are likely attributed to the higher level of literacy (94% literacy rate) [23], higher screening rates for diabetes [16], and higher burden of diabetes and its risk factors in Kerala [24,25] compared with other Indian states. The differences in time periods during which the studies were conducted and variations in population characteristics (e.g., knowledge rates of diabetes among urban residents, men, and those with higher educational status are likely to be higher than those of their counterparts) [7,8], across the studies may also have played a role.

Family history of diabetes was the most commonly reported risk factor for the disease. This is not surprising given that a family history of diabetes is highly prevalent in Kerala, with studies showing the proportions ranging from 47.9% in high-risk people for diabetes [26] to 73.8% in people with diabetes [25]. Large proportions (60–67%) of participants also felt that physical inactivity, unhealthy diet, and obesity could cause diabetes, which is reassuring. Interestingly, a reasonably large percentage (33.2–42.6%) of participants believed that alcohol and tobacco use are risk factors for diabetes. This may be related to the high rates of alcohol consumption and smoking in Kerala, particularly among men [27]. In a meta-analysis of five cohort studies with 114,287 adults, relative to never drinkers, current drinkers had no significantly elevated risk of type 2 diabetes at any level of drinking [28]. With regard to smoking, the results of a meta-analysis of 84 cohort studies with 5,853,952 participants showed that the pooled relative risk for diabetes incidence was 1.37 (95% CI 1.33–1.42) when comparing current smokers with nonsmokers [29].

There is strong evidence from the literature showing that diabetes causes micro- and macrovascular complications, mainly retinopathy and blindness, neuropathy, coronary heart disease, foot ulcers, and neuropathy [30]. In our study, the kidney was cited as the most commonly affected organ by type 2 diabetes. This is in contrast to other studies in India, where either the eyes [7,9] or feet [8] were more frequently reported by the general population or patients with diabetes. Worryingly, the proportion of our participants reporting other key organs such as the eyes, heart, feet, and nerves was very low. For example, only 24.0% and 10.2% believed that diabetes affects the eyes and feet, respectively. Studies show that diabetic retinopathy develops early in the natural course of diabetes before other complications become evident [30]. Diabetic foot ulcer is a serious diabetes-related complication that can lead to amputations and imposes an enormous economic burden on the individuals and healthcare system [31].

Our study has certain strengths and limitations. The internal consistency of the diabetes knowledge scale among our study population was high. In addition, the missing data to calculate the composite diabetes knowledge score were negligible (0.9%). The generalizability of our findings to other regions in the country is limited, as Kerala is a highly literate state in India [23] and has a higher prevalence of diabetes compared with other Indian states [32]. However, Kerala is in the most advanced stage of epidemiological transition [33] and is supposedly the "harbinger" for other Indian states in relation to the

burden of diabetes and other non-communicable diseases [24,34]. Thus, we can safely assume that what is happening now in Kerala concerning the knowledge of diabetes can occur in the rest of the country in the future. Since the prevalence of type 2 diabetes is low in Indians aged less than 30 years [35,36], we recruited individuals aged 30 years and above to the study. However, younger individuals (e.g., aged 20–30 years) are generally more concerned about health issues, such as being overweight or obese, as compared with older people. Thus, future studies on assessing the knowledge of diabetes should include younger individuals.

## 5. Conclusions

The overall level of knowledge of diabetes about its risk factors, complications, and prevention was generally higher than that reported by previous studies conducted among the general population in India. However, a very low proportion of our participants knew that diabetes can affect key organs such as the eyes, heart, feet, and nerves, and nearly a quarter were not aware that diabetes can be prevented. This is alarming given that these are high-risk individuals for diabetes, and the study was conducted in a state with the highest literacy rate (94.1%) [23] and the highest prevalence of diabetes (~20%) [32] in India. The healthcare system in the Trivandrum district of Kerala should, therefore, give more attention to educating high-risk individuals about diabetes complications and the importance of and strategies for diabetes prevention.

**Author Contributions:** Conceptualization: T.S.; methodology: T.S., K.R.T. and B.O.; formal analysis: T.S.; resources: K.R.T. and B.O.; data curation: T.S.; writing—original draft preparation: T.S.; writing—review and editing: T.S., K.R.T., J.P. and B.O.; supervision: B.O. All authors have read and agreed to the published version of the manuscript.

**Funding:** This research was supported by funding from the National Health and Medical Research Council, Australia for the K-DPP study (ID 1005324).

**Institutional Review Board Statement:** The K-DPP trial was approved by the Health Ministry Screening Committee of the Government of India; ethics committees of the Sree Chitra Tirunal Institute for Medical Sciences and Technology (SCT/IEC-333/May 2011), Trivandrum, India; Monash University (CF11/0457-2011000194); and The University of Melbourne (1441736) in Australia.

**Informed Consent Statement:** Written informed consent was obtained from all study participants.

**Data Availability Statement:** The data are available from the corresponding author upon reasonable request.

**Acknowledgments:** We thank the research staff for collecting the data for this study.

**Conflicts of Interest:** The authors declare no conflict of interest.

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
