# Peer review of "Knowledge of Diabetes among Adults at High Risk for Type 2 Diabetes in the Trivandrum District of Kerala, India"

_diabetology, doi:10.3390/diabetology4010009_

Round 1

Reviewer 1 Report

The manuscript is well-written with a clear rationale. However, the manuscript does not seem to contribute much to existing knowledge. Though the authors have addressed one of the crucial ongoing health issues in low and middle-income countries like India, they need to address some of the major questions.

p. 2, Line 45: The authors have mentioned that the study is about knowledge of diabetes among adults at high risk for T2DM in Kerala. The samples are collected from randomized areas in and around Trivandrum, the capital of Kerala. Authors have to elaborate if they have covered entire Kerala. Probably a schematic diagram would be more informative.

p. 2, Line 47: The reason for not choosing 20-30 years should be included (because this group of adults is more concerned about the health issues like obesity and being overweight).

p. 2, Line 49:  The rationale for the high-risk factor should be elaborated. Criteria for considering them as high-risk individuals.

p. 3, Line 86: The percentage in the graph (figure 1) is different from the text.

General Comments: Overall the grammar and written language should be edited. 

Reviewer 2 Report

Introduction is too general. Author should explain the study gap and epidemiology of diabetes in India and neighboring countries. What about type-1 diabetes? Why one type-2 diabetes is important? Please explain it in introduction clearly.

Author should give the other parameters i.e. Blood pressure and Kidney disease to make fruitful result, discussion and conclusion/

Author should explain the novelty of this study. How this study will be helpful for scientist/researcher, physician, and population?

Figure 1: Please write the X and Y-axis title clearly.

Line 86: 79.6%? please correct this.

Round 2

Reviewer 2 Report

The authors have addressed my comments /suggestions.